# Analysis of Chemical Composition and Antioxidant Activity of *Idesia polycarpa* Pulp Oil from Five Regions in China

**DOI:** 10.3390/foods12061251

**Published:** 2023-03-15

**Authors:** Wenlong Zhang, Chenwei Zhao, Emad Karrar, Meijun Du, Qingzhe Jin, Xingguo Wang

**Affiliations:** 1School of Food Science and Technology, Jiangnan University, Wuxi 214122, China; 2National Engineering Research Center for Functional Food, Jiangnan University, Wuxi 214122, China

**Keywords:** *Idesia polycarpa* pulp oil, chemical properties, antioxidant capacity

## Abstract

*Idesia polycarpa* pulp oil (IPPO) has the potential to become the new high-quality vegetable oil. The chemical parameters, fatty acid composition, bioactive ingredients, and antioxidant capacity of five Chinese regions of IPPO were studied comparatively, with significant differences among the regions. The oils were all abundant in unsaturated fatty acids, including linoleic acid (63.07 ± 0.03%–70.69 ± 0.02%), oleic acid (5.20 ± 0.01%–7.49 ± 0.03%), palmitoleic acid (4.31 ± 0.01%–8.19 ± 0.01%) and linolenic acid (0.84 ± 0.03%–1.34 ± 0.01%). IPPO is also rich in active substances such as tocopherols (595.05 ± 11.81–1490.20 ± 20.84 mg/kg), which are made up of α, β, γ and δ isomers, β-sitosterol (1539.83 ± 52.41–2498.17 ± 26.05 mg/kg) and polyphenols (106.77 ± 0.86–266.50 ± 2.04 mg GAE/kg oil). The free radical scavenging capacity of IPPO varies significantly depending on the region. This study may provide important guidance for the selection of *Idesia polycarpa* and offer insights into the industrial application of IPPO in China.

## 1. Introduction

Vegetable oils are widely employed in domestic cooking and the food business to satisfy dietary needs. The need for vegetable oils has risen sharply in recent years due to the growing worldwide population and rising living standards. To meet the demand, humans have made many efforts to increase the yields of oil crops or domesticate wild oilseed plants [1]. Most common vegetable oil plants are herbaceous and woody oil plants. Woody oil plants, which have a small footprint, have the advantage of their high oil output. In recent years, as the new woody oil plant, *Idesia polycarpa* has gradually caught the attention of researchers because of its high oil yield and nutritive values [2].

*Idesia polycarpa*, a deciduous tree of the Flacourtiaceae family, is native to China, Korea, Japan, and Russian Far East [3]. In China, *I. polycarpa* is mainly distributed south of the Qinling Mountains and the Huaihe River [4]. The plant was formerly used as an ornamental plant in China because of its strong adaptability and attractive appearance [5]. It is regarded as the “oil bank of trees” due to its high lipid content and oil yield, with a dry base oil content of 26.26% for the seeds and 43.6% for the pulp and an output of 2.25–3.75 tons of oil per hectare [6]. The oil extracted from *I. polycarpa* has been used as an edible oil in some provinces of China for hundreds of years and has proven safe for consumption based on toxicological tests and health surveys of people who consume it [2]. In addition, the leaves of the plant have been known to have hemostatic properties, the seeds have been employed as an insecticide, and the extraction of the fruits has the potential for anti-obesity [7,8].

*Idesia polycarpa* pulp oil (IPPO) is a high-quality oil extracted from the *Idesia polycarpa* pulp, which contains a higher concentration of linoleic acid than those of other woody oils, reaching 66–81% of the total fatty acids [9]. Previous studies have demonstrated that linoleic acid plays a significant role in fetal and infant development [10], hypertension [11], and diabetes [12]. In addition, IPPO is enriched with natural antioxidants and various active substances such as tocopherols, phytosterols and polyphenols. Due to its reasonable fatty acid composition and diverse minor substances, IPPO has the potential to become a high-quality edible oil.

However, as far as we know, previous studies focused on the extraction of fruit oil [13,14], and only a few studies have investigated the quality of pulp oil and seed oil. Some studies have already described the content of oil, fatty acid composition and lipid profile of different cultivars of *Idesia polycarpa* pulp oil and seed oil [15,16]. However, the studies related to the chemical characteristics and active ingredients from different regions of IPPO have not been systematically compared and analyzed. Furthermore, the free radical scavenging capabilities and quality of IPPO in different areas have not been investigated.

Therefore, the aims of this study were twofold: (i) to extract the oil in the pulp of *I. polycarpa* from different regions where the *I. polycarpa* were obtained from the major production locations in China; (ii) to investigate and contrast the chemical properties and bioactive ingredients and free radical scavenging capacity of the prepared IPPO. The results of this study can be used to guide the selection of *I. polycarpa* regions with outstanding quality.

## 2. Materials and Methods

### 2.1. Materials

Tocopherol standards, Mixed standards of 37 fatty acid methyl esters, gallic acid (99%), 5a-cholestanol standards, water-soluble tocopherol (Trolox), 2,2′-azino-bis(3-ethylbenzothiazoline-6-sulfonic acid) diammonium salt (ABTS), 2,4,6-tris(2-pyridyl)-s-triazine (TPTZ), 2,2′-azobis (2-methylpropionamidine) dihydrochloride (AAPH), 2,2-diphenyl-1-picrylhydrazyl (DPPH), fluorescein sodium salt (FL) were all purchased from Sigma-Aldrich Co., Ltd. (Shanghai, China). Chromatographic-grade methanol, ethanol, n-hexane, isopropanol and other analytical reagent grade reagents were from Sinopharm Medicine Holding Co., Ltd. (Shanghai, China).

Five regions of *I. polycarpa* were collected as materials in November 2022 from Xuancheng City (longitude: 104°55′, latitude: 25°48′, altitude: 450m), Xingyi City (longitude: 118°59′, latitude: 30°16′, altitude: 800 m), Ningqiang City (longitude: 106°15′, latitude: 32°50′, altitude: 875 m), Mianyang City (longitude: 104°45′, latitude: 31°47′, altitude: 500 m), and Liangshan City (longitude: 102°16′, latitude: 27°53′, altitude: 2200 m).

### 2.2. Preparation of IPPO

The fruits from five regions were in a deep red ripening period, and 200 g of *I. polycarpa* fruits from each region were selected as samples for oil extraction. The pulp and seed were separated from the collected *I. polycarpa* (the initial moisture content of the fruit was about 60%), and the pulp and seed were collected, then weighed to calculate pulp content. Then the pulp was dried in the oven (80 °C) for 8 h in darkness (the moisture content was about 3% after drying) and then crushed into a powder. Hexane was used as the extraction solvent, and the powder-to-solvent ratio was set at 1:5 (*w*:*v*). The lipid was extracted twice with fresh hexane for two hours at room temperature. The collected solvent was then extracted by suction in a 50 °C rotating evaporator until it totally evaporated. The IPPO obtained was kept in brown glass jars in darkness at 4 °C.

### 2.3. Oil Content and Chemical Properties

The oil content of the *I. polycarpa* pulp was determined based on the methods described by AOCS Am 2-93 [17]. The acid value, peroxide value, iodine value and saponification value of the IPPO were determined by the American Oil Chemists Society (AOCS) methods (Cd 3d-63, Cd 8b-90, Cd 1d-92 and Cd 3-25, respectively) [17].

### 2.4. Fatty Acid Composition

Fatty acid methyl esters (FAMEs) were made in accordance with Shi’s method [18] with slight modifications. A sample of oil (50 mg) was mixed with 2 mL n-hexane and 1 mL methanolic potassium hydroxide (2 mol/L). The mixture was shaken for 1 min, kept in a water bath for 30 min at 50 °C and added 1 mL saturated sodium chloride solution before cooling and collected the supernatant after centrifugation. The composition of fatty acids was then determined using a gas chromatography system (Agilent 7820A, Agilent Technologies, Inc., Santa Clara, CA, USA) equipped with a capillary column (TRACE TR-FAME, 60 m × 250 μm, 0.25 μm, Thermo Fisher Scientific, Waltham, MA, USA) and a flame ionization detector. The heating procedure was carried out at 80 °C for 0.5 min, then to 165 °C at 40 °C/min and held for 1 min, then increased to 230 °C at 2 °C/min and held at 230 °C for 2 min. Helium was used as the carrier gas in a steady flow of 0.8 mL/min, and the injection volume was 1 μL with an autosampler In split mode (the split ratio was set at 10:1). The temperatures of the injector, transfer line and ion source were set at 260, 260 and 250 °C, respectively. The targeted FAME were determined by comparing their retention times with the standards, and the content was expressed as the relative percentage of the total area.

### 2.5. Tocopherols Content

The method for determining tocopherol content was described in Wang [19] with modifications. Each oil sample (1 g) was diluted with n-hexane into a volume in a 10 mL volumetric flask. The mixture was then filtered with a 0.22 μm nylon syringe filter, and 20 μL of the solution was injected into a Waters 2996 HPLC for analysis. The tocopherols were measured at 295 nm using a liquid chromatography system (Waters-1525, Waters, Inc., Milford, MA, USA) equipped with a PDA detector and a Sehperisorb SiO_2_ column (250 mm × 4.6 mm, 5 μm, Hanbon, Jiangsu, China) at a flow rate of 0.8 mL/min with a mobile phase of hexane: isopropanol (98:2, *v*/*v*). The column temperature was set at 40 °C. The fitting curves were shown in Appendix A.

### 2.6. β-Sitosterol Content

The determination of β-sitosterol content was based on Shi [18]. 300 mg of oils were weighed in the sample bottle, 0.5 mL of 0.5 mg/mL 5α-cholestane (dissolved in n-hexane) and 2 mL of 2 mol/L KOH-Ethanol solution were added. The mixture was heated to 85 °C for 1 h and then cooled to room temperature, upon which 2 mL of distilled water and 5 mL of n-hexane were added. The top layer was extracted twice with 5 mL of n-hexane, and the extracts were collected together and dried with nitrogen. Then the product was silylated with 0.2 mL of BSTFA + TMCS for 30 min at 75 °C, and the mixture was then cooled and filtered through a 0.22 μm nylon syringe filter. The gas chromatography-mass system was equipped with a capillary column (DB-5MS, 30 m × 0.25 m, 0.25 µm; Agilent Technologies, Inc., Santa Clara, CA, USA) and a flame ionization detector (Thermo Fisher, Inc., Waltham, MA, USA). The heating procedure was carried out at 200 °C for 0.5 min, then increased to 300 °C at 10 °C/min and held at 300 °C for 18 min, resulting in a detector and injector temperature of 280 °C and a program run time of 28.5 min. Helium was used as the carrier gas in a constant flow of 1.5 mL/min, and the injection volume was 1 μL with an autosampler in split mode (the split ratio was set at 100:1). The temperature of the ion source was set at 250 °C. β-sitosterol was identified by comparing the relative retention time from the oil samples with that obtained from the standards. Quantification was carried out using the internal standard 5a-cholestane.

### 2.7. Total Polyphenol

The total polyphenol content of IPPO was analyzed according to Karrar’s method [20]. The total polyphenol was extracted by Sepax Generik Diol tube (Sepax Technologies, Inc., Newark, DE, USA). The absorbance was measured at 765 nm after a 2 h reaction in darkness. The amount of total polyphenol content was calculated by plotting a calibration curve with gallic acid as a reference standard. The results were expressed in milligrams of gallic acid equivalent (mg GAE/kg oil) consumed per kg sample. The fitting curve was shown in Appendix A.

### 2.8. Free Radical Scavenging Capacity

The free radical scavenging ability of the polar fraction of IPPO was evaluated using the method of Shi [18] with improvements: 4.0 g of sample was mixed with 5 mL of methanol for 30 min in the dark, then centrifuged at 3000 r/min for 15 min, after complete separation of vegetable oil and methanol, the supernatant was extracted 4 times in succession as described above, and the extracts were combined in a 20 mL brown sample bottle and stored at −20 °C for further analyses. The absorbance signal was translated to antioxidant activity using Trolox (dissolved in methanol) as the reference standard. The result was expressed as μmol Trolox equivalent per 100 g of sample (μmol TE/100 g of oil).

#### 2.8.1. ABTS Assay

The ABTS radical scavenging ability of IPPO was measured using the evaluation method of Cai [21] with modifications. 25 mL of ABTS reagent (7 mmol/L) was mixed with 440 μL of potassium persulphate solution (2.45 mmol/L) and placed in darkness for 12–16 h. The prepared ABTS working solution was diluted with methanol to have an absorbance of 0.7 ± 0.02 at 734 nm before use. The final reaction mixture was made up of 0.25 mL of polar extract and 5 mL of ABTS working solution. The absorbance was determined at 734 nm using a spectrophotometer after the mixture had been shaken for one minute and left at room temperature in the dark for 20 min. The recorded absorbance signal was converted to antioxidant activity using Trolox as the reference standard.

#### 2.8.2. DPPH Assay

The DPPH radical scavenging capability of IPPO was calculated by referring to the assay of Al Juhaimi [22] with modifications. The DPPH methanol solution (2 mL) was mixed with 1 mL of polar extract, and then the mixture was left in the dark for two hours. The absorbance at 517 nm was analyzed with a spectrophotometer.

#### 2.8.3. FRAP Assay

The method for determining IPPO’s FRAP radical scavenging capacity was based on Szydłowska-Czerniak [23] and improved. A 10 mmol/L TPTZ solution was first made with 40 mmol/L hydrochloric acids. 10 mL of FeCl_3_ solution (20 mmol/L), 10 mL of TPTZ solution (10 mmol/L), and 100 mL of acetate buffer (0.1 mol/L, pH 3.6) were mixed in a brown bottle, placed at a constant temperature of 37 °C and used as needed. Then 3 mL of FRAP reagent and 0.1 mL of polar extract were added to a 10 mL brown volumetric flask and fixed with distilled water. The extract was centrifuged at 5000 rpm for 10 min after standing for 10 min, and the absorbance at 593 nm was measured with a spectrophotometer.

#### 2.8.4. ORAC Assay

The ORAC radical scavenging capacity of IPPO was determined by referencing the method of Szydłowska-Czerniak [23] and modified. A solution of 8.16 × 10^−5^ mmol/L FL and 153 mmol/L AAPH in 75 mmol/L phosphate buffer (pH = 7.4) was prepared, and 150 μL of FL solution and 25 μL of methanol extract was added to a clear 96-well black plate. The mixture was shaken for 2 min and then stored in the dark at 37 °C for 10 min, and 25 μL AAPH solution was added to initiate the reaction. Fluorescence degradation was measured every 5 min for 5 h at 37 °C using a fluorescence spectrophotometer at 525 nm emission and 485 nm excitation wavelength. The natural fluorescence decay of the FL solution and methanol blank were used as controls.

### 2.9. Data Analysis

Each sample was determined in triplicate, and the results were presented as mean ± standard deviation (SD). Using SPSS 21.0 Statistical software (IBM SPSS Inc., Armonk, NY, USA), data were compared using Duncan’s multiple range test at *p* < 0.05. Principal component analysis (PCA) was performed with Origin 2021 (Origin Lab Inc., Northampton, MA, USA).

## 3. Results and Discussion

### 3.1. Pulp Content, Oil Content, and Chemical Parameters

The pulp content of *I. polycarpa*, the oil content of *I. polycarpa* pulp, and the chemical parameters of IPPO tested are shown in Table 1. The pulp was the most important portion of the *I. polycarpa*, with relative percentages ranging from 67.71 ± 1.32% to 74.71 ± 1.46%. The *I. polycarpa* of the Ningqiang cultivar had the highest pulp content, significantly higher than that of the Mianyang cultivar, which had a pulp content of only 67.71 ± 1.32%.

The oil content of *I. polycarpa* pulp ranged from 30.48 ± 0.19% to 40.09 ± 0.07%; significant differences were discovered in the different regions (*p* < 0.05). Liangshan cultivar had the highest oil content of 40.09%, followed by Ningqiang (37.53 ± 0.39%), Xingyi (36.22 ± 0.10%), Mianyang (33.04 ± 0.20%) and Xuancheng (30.48 ± 0.19%), the oil content was higher than the IPPO reported by Yang (26.26%) [6], closed to the amount reported by Li(20–40%) [24]. In addition, the oil content of *I. polycarpa* pulp was higher than soybean (25.6%) [25], safflower seed (25.1%) [26], ripe olives (17.50–20.25%) [27] and lower than palm fruit peel (46.02%) [28], which demonstrated significant prospects as a new woody oil resource with high commercial value. The oil content of IPPO is substantially associated with the altitude of its regions.

According to Table 1, the chemical parameters of IPPO, including acid, peroxide, iodine, and saponification values, differed greatly depending on the region. The acid values of IPPO ranged from 0.98 ± 0.02 to 2.25 ± 0.02 mg KOH/g, which was lower than that of sesame oil (3.78 mg KOH/g) [29] and walnut oil (2.49 mg KOH/g) [30]. The low acid value indicated that IPPO had strong storage stability, low oil rancidity, and good quality, making it suitable as an edible oil.

Peroxide value can reflect the content of peroxides and hydroperoxide that were produced during the first stage of fat oxidation [31]. The peroxide values of IPPO ranged from 0.48 ± 0.01 to 9.40 ± 0.03 mmol/kg, demonstrating a significant effect of the cultivar. Except for the Xuancheng cultivar, the peroxide value of the other cultivars was lower than that of torreya seed oil (1.69–3.43 mmol/kg) [32], olive oil (3.20 mmol/kg) and rapeseed oil (4.73 mmol/kg) [33], but lower than that of palm oil (7.98 mmol/kg) [34].

The iodine value indicated the fatty acid unsaturation [18]. The iodine values of IPPO were 123.04 ± 1.60–151.07 ± 0.69 gI_2_/100 g, which was higher than that of peanut oil (111.19 gI_2_/100g) and olive oil (80.03 gI_2_/100 g) [33], which indicated that IPPO had lots of unsaturated fatty acid.

The saponification value reflected the content of glyceride and indicated the level of companion [18]. The saponification values of IPPO ranged from 164.70 ± 2.27 to 185.45 ± 1.17 mg KOH/g, which indicated that the content of the companion was high. Xingyi cultivar had the highest saponification value (185.45 ± 1.17 mg KOH/g), followed by the Ningqiang cultivar (182.61 ± 1.14 mg KOH/g), Xuancheng cultivar (181.91 ± 0.72 mg KOH/g) and Liangshan cultivar (165.23 ± 1.51 mg KOH/g). Mianyang cultivar had the lowest cultivar (164.70 ± 2.27 mg KOH/g). The saponification value of IPPO was comparable to that of torreya seed oil (181.04–195.74 mg KOH/g) [35], soybean oil (179.45 mg KOH/g) [36], and palm oil (120.62 ± 1.40–247.76 ± 2.14 mg KOH/g) [37]. It indicated that triacylglycerol in IPPO contained relatively more short-chain fatty acids [35].

### 3.2. Fatty Acid Composition

Table 2 and Appendix A illustrates the composition of fatty acids from five regions of IPPO and identifies six main fatty acids and three others. The major fatty acids include palmitic acid (C16:0), palmitoleic acid (C16:1), stearic acid (C18:0), oleic acid (C18:1n9c), linoleic acid (C18:2), α-linolenic acid (C18:3). Linoleic acid (C18:2) was the main fatty acid of IPPO, with the percentages ranging from 63.07 ± 0.03% to 70.69 ± 0.02%, followed by palmitic acid (C16:0), oleic acid (C18:1), and palmitoleic acid (C16:1), with the contents of 14.19 ± 0.01%–19.55 ± 0.01%, 5.20 ± 0.01%–7.49 ± 0.03% and 4.31 ± 0.01%–8.19 ± 0.01%, respectively. The least prevalent compounds were linolenic acid (C18:3) and stearic acid (C18:0), which had concentrations of 0.84 ± 0.03%–1.34 ± 0.01% and 1.01 ± 0.01%–1.93 ± 0.01%, respectively. The other fatty acids were Heptadecanoic acid (C17:0), Arachidic acid (C20:0), and Heneicosanoic acid (C21:0), with percentages of 0.09 ± 0.01%–0.40 ± 0.00%, 0.33 ± 0.01%–0.77 ± 0.01% and 0.27 ± 0.01%–0.93 ± 0.02%, respectively. The five kinds of IPPO varied greatly in their relative proportion of these fatty acids.

Liangshan cultivar had the highest linoleic acid content (70.69 ± 0.02%), which was higher than soybean oil (50.17 ± 0.83%) [36], peanut oil (23.69 ± 0.03%) and olive oil (8.50 ± 0.11%) [33] but almost lower than a few oils such as safflower seed oil (80%) [38], closed to walnut oil (46.9–68.6%) [39]. The Mianyang cultivar had the lowest linoleic acid content (63.07 ± 0.03%), the Xingyi cultivar had the highest palmitic acid content (19.55 ± 0.01%), which was also higher than soybean oil (15.65 ± 0.03%) [36], olive oil (15.11 ± 0.36%) [33] and other common edible oils. Xuancheng cultivar also had the highest content of palmitoleic acid (8.19 ± 0.01%), distributed in most vegetable oils. Still, common vegetable oils such as olive oil (1.24 ± 0.03%) and rapeseed oil (0.17 ± 0.02%) [33] had lower content, and the content of palmitoleic acid in IPPO was unusual in vegetable oils. Compared to the common edible oils, IPPO had low concentrations of oleic acid, stearic acid, and linolenic acid. The fatty acid contents in IPPO were similar to those reported by Yang [6] and Li [24]. Linoleic acid is an important fatty acid that benefits the healthy growth of human skin [40]. It helps to prevent cardiovascular disease by reducing plasma cholesterol and changing its distribution in the body [41]. Palmitoleic acid can reduce inflammation, prevent diabetes and cardiovascular diseases, and has application potential in skin whitening [42]. Linoleic acid is a necessary fatty acid to maintain some key physiological functions of the human body [33]. IPPO contains high levels of functional fatty acids and can potentially benefit consumers as a source of nutrition.

Moreover, the oil had a high unsaturated fatty acid/saturated fatty acid (UFA/SFA) ratio, which is beneficial in reducing serum cholesterol and atherosclerosis, and also is good at preventing heart disease [43]. The UFA/SFA ratio of the IPPO was 3.43 ± 0.00–4.96 ± 0.01. The Liangshan and Mianyang cultivars had the highest ratios (4.96 ± 0.01 and 4.43 ± 1.00, respectively), followed by the Ningqiang cultivar (4.25 ± 0.00), and the Xuancheng and Xingyi cultivars had the lowest ratios (3.77 ± 0.01 and 3.43 ± 0.00, respectively). The rations of IPPO were similar to those of soybean oil (3.69) and olive oil (3.026–5.826) [36,44]. This indicated that IPPO had the potential to be used as edible oils in the cooking and food industry.

### 3.3. Tocopherol Content

Tocopherol is an important antioxidant in IPPO. We used HPLC to identify α, β, γ and δ isomers (Appendix A). The composition and content of tocopherol compounds among different regions are shown in Table 3. The main tocopherol in IPPO was α-tocopherol with a content of 269.51 ± 8.71–828.01 ± 15.22 mg/kg. The amounts of β, γ, and δ-tocopherols were 54.99 ± 5.4–103.60 ± 0.85 mg/kg, 10.68 ± 1.08–374.34 ± 14.93 mg/kg and 25.98 ± 8.42–207.28 ± 11.55 mg/kg, respectively. The α, γ, and δ-tocopherol concentrations of the Xingyi cultivar were the highest and much higher than other cultivars; Liangshan had the highest β-tocopherol content. IPPO contained 595.05 ± 11.81–1490.20 ± 20.84 mg/kg of tocopherols, higher than the majority of popular edible vegetable oils such as sunflower oil, rapeseed oil, olive oil, sesame oil and soybean oil (141.3–785.1 mg/kg) [45].

Tocopherols can enhance the stability of edible oils and have anti-inflammatory and anti-proliferative capabilities [46]. Taken together, IPPO was rich in tocopherols and dominated by α-tocopherol, which indicated that IPPO had high nutritional value and may have better oxidative stability and free radical scavenging ability.

### 3.4. β-Sitosterol Content

β-sitosterol is a key component of phytosterols which is useful in lowering hypercholesterolemia [47]. Total ion chromatogram of β-sitosterol was shown in Appendix A. The β-sitosterol content of IPPO ranged from 1539.83 ± 52.41 to 2498.17 ± 26.05 mg/kg. The Xingyi cultivar had the highest content, followed by the Ningqiang cultivar (2413.08 ± 33.17 mg/kg), Xuancheng cultivar (2049.3 ± 40.52 mg/kg), Liangshan cultivar (1631.67 ± 29.37 mg/kg), and Mianyang cultivar (1539.83 ± 52.41 mg/kg) was the lowest. In comparison to, soybean oil (1238.9 mg/kg), coconut oil (385.0 mg/kg), camellia seed oil (377.7 mg/kg), flax oil (155.41 mg/kg), palm oil (357.2 mg/kg), and sunflower oil (153.43 mg/kg) [33], IPPO had a substantially greater concentration of β-sitosterol. Thus, it suggested that IPPO, rich in β-sitosterol, may provide potential health benefits.

### 3.5. Total Polyphenol Content

Phenolic compounds are crucial trace molecules found in vegetable oils, which are responsible for both the sensory and nutritional properties of these oils [46]. The polyphenol content of IPPO had a significant difference among five regions (*p* < 0.05), which ranged from 106.77 ± 0.86 to 266.50 ± 2.04 mg GAE/kg oil, and higher than rapeseed oil, wheat germ oil (57.7 ± 1.3, 25.2 ± 0.4 mg GAE/kg oil), rice bran oil, walnut oil, linseed oil, and peanut oil (21.8 ± 1.6, 19.0 ± 0.9, 20.9 ± 0.7, 18.4 ± 0.6 mg GAE/kg oil) [45]. Ningqiang cultivar had the highest content of polyphenol, followed by the Liangshan cultivar (188.11 ± 1.89 mg GAE/kg oil), the Xuancheng cultivar (169.65 ± 0.97 mg GAE/kg oil), Xingyi cultivar (156.38 ± 0.62 mg GAE/kg oil) and Mianyang cultivar (106.77 ± 0.86 mg GAE/kg oil). In addition, the presence of phenolic components in oil aids in protecting other minor components from oxidation, such as tocopherols [48].

### 3.6. Free Radical Scavenging Ability

The free radical scavenging ability of vegetable oil reflects the nutritional and health care functions of vegetable oil to some extent. It can be used as an index to evaluate oil quality comprehensively. The free radical scavenging capacity of the polar portion of five distinct regions of the IPPO was assessed using the ORAC, ABTS, FRAP, and DPPH assessment methodologies. The results were expressed as μmol TE/100 g oil; The ORAC method characterizes the ability of the antioxidant substances in the sample to absorb free radicals. The antioxidant components in the sample enable the ABTS technique to convert the blue-green radical cation ABTS+ to a colorless ABTS molecule [49]; The decrease of Fe^3+^-TPTZ to a blue-violet color establishes FRAP’s TAC value. The DPPH method is measured by the change in absorbance value caused by the reaction of DPPH with the antioxidant substance in the sample.

From Table 4, it is known that the polar fractions of six different cultivars of IPPO had the following effects on the ABTS method, the DPPH method, FRAP method, and ORAC method were 177.82 ± 4.74–387.88 ± 2.37 μmol TE/100 g of oil, 149.30 ± 6.49–288.01 ± 3.74 μmol TE/100 g of oil, 133.13 ± 1.56–280.94 ± 1.88 μmol TE/100 g of oil, 95.77 ± 9.60–212.37 ± 0.16 μmol TE/100 g oil with the strongest free radical scavenging ability for ABTS. The Ningqiang cultivar had the best capacity to scavenge DPPH* and ABTS* + free radicals, while the Xingyi cultivar had the strongest ability to scavenge FRAP and ORAC free radicals. Both cultivars showed a good ability to prevent the formation of free radicals.

### 3.7. Principal Component Analysis

PCA was used to characterize the oil samples according to their chemical property, oil content, fatty acid compositions, β-sitosterol, total tocopherol, polyphenol, and free radical scavenging capacity. The PCA results showed two major components accounted for 74.4% of the data variance (PC1: 44.3%, PC2: 30.1%). Figure 1a shows the vectors of the variables and their contributions to the first two principal components. It could be seen that the first principal component was positively related to oil content, oleic acid, linoleic acid, unsaturated fatty acid and polyphenol, however, was negatively related to peroxide value, acid value, palmitic acid, palmitoleic acid and saturated fatty acid. The β-sitosterol, total tocopherol and free radical scavenging capacity had positive loadings on PC2, whereas unsaturated fatty acid and oil content possessed positive loadings on PC2.

Figure 1b clearly shows that the IPPO from five regions could be well distinguished based on the quality of oils. Liangshan cultivar was characterized by high levels of oleic acid, linoleic acid, unsaturated fatty acid, low acid value and peroxide value. Ningqiang cultivar had a high polyphenol content and moderate content of other active substances. Xingyi cultivar was characterized by a high percentage of β-sitosterol and total tocopherol. Compared with other cultivars, the antioxidant activity of the Xingyi cultivar was high. Xuancheng cultivar and Mianyang cultivar were gathered into the same group, which had high levels of palmitic acid, palmitoleic acid, saturated fatty acid, high acid value and peroxide value, low β-sitosterol, total tocopherol and polyphenol concentrations.

On the whole, the chemical properties, fatty acid composition, active substances and antioxidant activity of the IPPO from five regions were evaluated, and differences among the cultivars were also observed. According to the comprehensive index, the Liangshan cultivar, Ningqiang cultivar and Xingyi cultivar had outstanding qualities, such as high levels of oleic acid, linoleic acid, high content of β-sitosterol, total tocopherol and polyphenol and high antioxidant activity. The results may provide help for the selection of IPPO with specific quality characteristics for the food industry.

## 4. Conclusions

The present study demonstrates that the five regions of IPPO differ significantly in their chemical properties and antioxidant capacity, suggesting that this finding may be significant for selecting high-value IPPO. Linoleic acid was the predominant fatty acid of IPPO, ranging from 57.31 ± 0.01% to 70.68 ± 0.03%. In addition, IPPO had a higher palmitoleic acid content than common edible oils. Depending on the region, the content of tocopherols, β-sitosterol, polyphenols, and other trace chemicals in IPPO and its ability to scavenge free radicals varied significantly. PCA results showed that the Xingyi cultivar had the highest α-tocopherols, β-sitosterol content, FRAP, and ORAC free radical scavenging capacity, the Ningqiang variety had the highest total phenol content, ABTS and DPPH free radical scavenging capacity, Liangshan cultivar had highest oil content and linoleic acid content. These findings showed that IPPO was a promising choice for edible oils or functional components in the food industry since it had a high nutritional value and tremendous health benefits.

## Figures and Tables

**Figure 1 foods-12-01251-f001:**
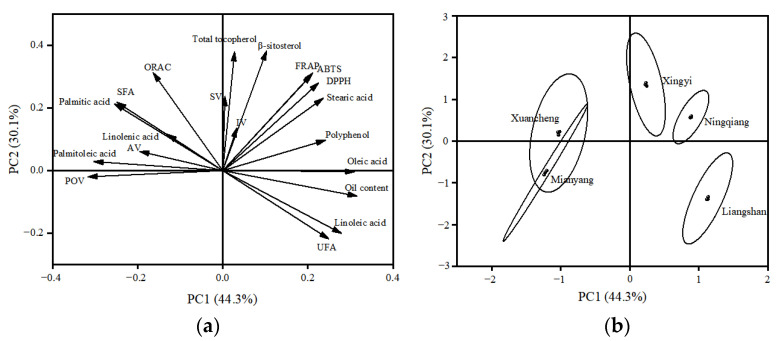
Multivariate general characterization of IPPO from five regions by PCA. Plots of coefficient (**a**), and cultivar (**b**).

**Table 1 foods-12-01251-t001:** Pulp content of *I. polycarpa*, oil content of *I. polycarpa* pulp, and chemical properties of IPPO from five regions.

	Ningqiang	Xuancheng	Xingyi	Mianyang	Liangshan
Pulp content (%)	74.71 ± 1.46 ^a^	69.28 ± 2.55 ^b^	68.79 ± 1.50 ^b^	67.71 ± 1.32 ^b^	73.54 ± 1.88 ^a^
Oil content (%)	37.53 ± 0.39 ^b^	30.48 ± 0.19 ^e^	36.22 ± 0.10 ^c^	33.04 ± 0.20 ^d^	40.09 ± 0.07 ^a^
Acid value (mg KOH/g)	1.77 ± 0.01 ^b^	2.25 ± 0.02 ^a^	1.04 ± 0.02 ^d^	1.58 ± 0.02 ^c^	0.98 ± 0.02 ^e^
Peroxide value (mmol/kg)	2.47 ± 0.24 ^c^	9.40 ± 0.03 ^a^	1.83 ± 0.08 ^d^	7.83 ± 0.10 ^b^	0.48 ± 0.01 ^e^
Iodine value (gI_2_/100g)	137.43 ± 1.04 ^b^	151.07 ± 0.69 ^a^	124.60 ± 0.38 ^c^	123.04 ± 1.60 ^c^	125.27 ± 1.44 ^c^
Saponification value (mg KOH/g)	182.61 ± 1.14 ^ab^	181.91 ± 0.72 ^b^	185.45 ± 1.17 ^a^	164.70 ± 2.27 ^c^	165.23 ± 1.51 ^c^

Each value is expressed as the mean ± standard deviation (*n* = 3). Superscripts with different letter in the same row was statistically significant (*p* < 0.05).

**Table 2 foods-12-01251-t002:** Fatty acid composition of IPPO from five regions.

Fatty Acid (%)	Ningqiang	Xuancheng	Xingyi	Mianyang	Liangshan
Palmitic acid (C16: 0)	16.18 ± 0.00 ^d^	18.19 ± 0.02 ^c^	19.55 ± 0.01 ^a^	19.48 ± 0.01 ^b^	14.19 ± 0.01 ^e^
Palmitoleic acid (C16: 1)	5.98 ± 0.01 ^c^	8.19 ± 0.01 ^a^	5.21 ± 0.02 ^d^	7.57 ± 0.03 ^b^	4.31 ± 0.01 ^e^
Heptadecanoic acid (C17: 0)	0.40 ± 0.00 ^a^	0.39 ± 0.01 ^b^	0.16 ± 0.00 ^d^	0.27 ± 0.01 ^c^	0.09 ± 0.01 ^e^
Stearic acid (C18: 0)	1.59 ± 0.01 ^b^	1.01 ± 0.01 ^e^	1.93 ± 0.01 ^a^	1.07 ± 0.01 ^d^	1.45 ± 0.02 ^c^
Oleic acid (C18: 1)	7.49 ± 0.03 ^a^	5.20 ± 0.01 ^e^	6.64 ± 0.02 ^c^	5.94 ± 0.02 ^d^	7.20 ± 0.02 ^b^
Linoleic acid (C18: 2)	66.63 ± 0.03 ^b^	64.29 ± 0.03 ^d^	64.36 ± 0.03 ^c^	63.07 ± 0.03 ^e^	70.69 ± 0.02 ^a^
Linolenic acid (C18: 3)	0.84 ± 0.03 ^e^	1.34 ± 0.01 ^a^	1.20 ± 0.01 ^b^	0.94 ± 0.01 ^d^	1.03 ± 0.04 ^c^
Arachidic acid (C20: 0)	0.48 ± 0.01 ^d^	0.60 ± 0.01 ^c^	0.33 ± 0.01 ^e^	0.73 ± 0.01 ^b^	0.77 ± 0.01 ^a^
Heneicosanoic acid (C21: 0)	0.41 ± 0.02 ^d^	0.78 ± 0.01 ^b^	0.61 ± 0.02 ^c^	0.93 ± 0.02 ^a^	0.27 ± 0.01 ^e^
SFA	19.05 ± 0.01 ^d^	20.96 ± 0.03 ^c^	22.58 ± 0.03 ^a^	22.48 ± 0.03 ^b^	16.78 ± 0.03 ^e^
UFA	80.95 ± 0.02 ^b^	79.03 ± 0.03 ^c^	77.42 ± 0.02 ^e^	77.52 ± 0.04 ^d^	83.23 ± 0.04 ^a^
MUFA	13.47 ± 0.02 ^b^	13.40 ± 0.01 ^c^	11.85 ± 0.03 ^d^	13.51 ± 0.01 ^a^	11.50 ± 0.02 ^e^
PUFA	67.48 ± 0.02 ^b^	65.64 ± 0.04 ^c^	65.57 ± 0.02 ^d^	64.01 ± 0.03 ^e^	71.72 ± 0.03 ^a^
UFA/SFA	4.25 ± 0.00 ^abc^	3.77 ± 0.01 ^bc^	3.43 ± 0.00 ^c^	4.43 ± 1.00 ^ab^	4.96 ± 0.01 ^a^

Each value is expressed as the mean ± standard deviation (*n* = 3). Superscripts with different letter in the same row was statistically significant (*p* < 0.05). SFA, saturated fatty acid; UFA, unsaturated fatty acid; MUFA, monounsaturated fatty acid; PUFA, polyunsaturated fatty acid.

**Table 3 foods-12-01251-t003:** Contents of tocopherols, β-sitosterol, and polyphenols in IPPO of five regions.

	Ningqiang	Xuancheng	Xingyi	Mianyang	Liangshan
α-tocopherol (mg/kg)	269.51 ± 8.71 ^d^	647.40 ± 18.03 ^b^	828.01 ± 15.22 ^a^	397.66 ± 16.42 ^c^	374.49 ± 6.67 ^c^
β-tocopherol (mg/kg)	59.73 ± 1.62 ^c^	57.95 ± 3.90 ^c^	80.56 ± 7.21 ^b^	54.99 ± 5.48 ^c^	103.60 ± 0.85 ^a^
γ-tocopherol (mg/kg)	115.31 ± 1.05 ^c^	10.68 ± 1.08 ^d^	374.34 ± 14.93 ^a^	182.57 ± 1.91 ^b^	111.12 ± 7.53 ^c^
δ-tocopherol (mg/kg)	150.49 ± 0.93 ^b^	25.98 ± 8.42 ^d^	207.28 ± 11.55 ^a^	138.79 ± 3.42 ^bc^	126.99 ± 2.67 ^c^
Total tocopherol (mg/kg)	595.05 ± 11.81 ^d^	742.02 ± 14.63 ^b^	1490.20 ± 20.84 ^a^	774.01 ± 23.89 ^b^	716.20 ± 15.05 ^c^
β-sitosterol (mg/kg)	2413.08 ± 33.17 ^a^	2049.3 ± 40.52 ^b^	2498.17 ± 26.05 ^a^	1539.83 ± 52.41 ^c^	1631.67 ± 29.37 ^c^
Polyphenol (mg GAE/kg oil)	266.50 ± 2.04 ^a^	169.65 ± 0.97 ^c^	156.38 ± 0.62 ^d^	106.77 ± 0.86 ^e^	188.11 ± 1.89 ^b^

Each value is expressed as the mean ± standard deviation (*n* = 3). Superscripts with different letter in the same row was statistically significant (*p* < 0.05).

**Table 4 foods-12-01251-t004:** Free radical scavenging ability of five regions of IPPO.

	Ningqiang	Xuancheng	Xingyi	Mianyang	Liangshan
ABTS (μmol TE/100 g)	387.88 ± 2.37 ^a^	240.45 ± 5.95 ^c^	353.19 ± 5.95 ^b^	177.82 ± 4.74 ^d^	241.23 ± 2.37 ^c^
DPPH (μmol TE/100 g)	288.01 ± 3.74 ^a^	227.62 ± 8.65 ^c^	278.10 ± 1.87 ^b^	149.30 ± 6.49 ^d^	226.44 ± 0.71 ^c^
FRAP (μmol TE/100 g)	259.38 ± 7.81 ^b^	194.06 ± 2.5 ^c^	280.94 ± 1.88 ^a^	133.13 ± 1.56 ^d^	199.69 ± 1.25 ^c^
ORAC(μmol TE/100 g)	204.24 ± 1.63 ^ab^	190.22 ± 6.78 ^c^	212.37 ± 0.16 ^a^	200.67 ± 5.13 ^bc^	95.77 ± 9.60 ^d^

Each value is expressed as the mean ± standard deviation (*n* = 3). Superscripts with different letter in the same row was statistically significant (*p* < 0.05). ABTS, 2,2′-diazobis (3-ethyl benzothiazoline -6-sulfonic acid) diammonium salt free radical scavenging capacity; DPPH, 2,2-diphenyl-1-trinitrohydrazine free radical scavenging capacity; FRAP, reduction antioxidant potential of ferric iron; ORAC, oxygen free radical absorption capacity.

## Data Availability

Data is contained within the article.

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
