# Peer review of "Analysis of Chemical Composition and Antioxidant Activity of Idesia polycarpa Pulp Oil from Five Regions in China"

_foods, 2023, doi:10.3390/foods12061251_

Round 1
Reviewer 1 Report
The manuscript “Analysis of chemical composition and antioxidant activity of Idesia polycarpa pulp oil from five regions in China” was focused on physicochemical properties, bioactive ingredients and free radical scavenging capacity of oil extracted from Idesia polycarpa pulp. Through reviewing the manuscript, I have several following minor points:
- Would the authors add representative chromatograms showing fatty acids, sitosterol, and tocopherols?
- It would be great if principal component analysis was used as it is a mathematical tool to be able to differentiate the samples from different regions. This would help highlight the similarities and dissimilarities among the samples.
Reviewer 2 Report
Reviewing of Analysis of chemical composition and antioxidant activity of Idesia polycarpa pulp oil from five regions in China
Reference. ID: foods-2204733
Overall appreciation
The paper deals with Analysis of chemical composition: fatty acids, antioxidants Contents of tocopherols, β-sitosterol, polyphenols and antioxidant properties of Idesia polycarpa pulp oil from five regions in China.
The main concern of this work is that: the main component fatty acids seem to be qualitative analyzed peak Area are presented and not % of total oil. Extraction method should be more clarified if Folch extraction method is used,
Besides for better analysis of all results, PCA analysis oh heat mapping could be applied to all generated data to better understand variabilities among regions,
Abstract:
-Lines 20-21, the word breeding is not appropriate.
“for the breeding of Idesia polycarpa and offer insights into the industrial application of IPPO in China.”
Introduction
-Lines 52-54, novelty to be more justified considering the following refs. The fruit is = pulp + seed, so please explain the interest to investigate both fruit parts separately.
Please add the following references in your intro and ref list:
1. Composition, characteristics and antioxidant activities of fruit oils from Idesia polycarpa using homogenate-circulating ultrasound-assisted aqueous enzymatic extraction. Author links open overlay panelKexin Hou a b, Xuebing Yang c, Meili Bao b, Fengli Chen a, Hao Tian d, Lei Yang a Industrial Crops and ProductsVolume 117, July 2018, Pages 205-215
2. High-pressure supercritical carbon dioxide extraction of Idesia polycarpa oil: Evaluation the influence of process parameters on the extraction yield and oil quality. Author links open overlay panelDan Zhou a 1, Xue Zhou a 1, Qinglong Shi a, Jiangbo Pan b, Huashu Zhan b, Fahuan Ge a Industrial Crops and ProductsVolume 188, Part A, 15 November 2022, 115586
3. Composition, characteristics and antioxidant activities of fruit oils from Idesia polycarpa using homogenate-circulating ultrasound-assisted aqueous enzymatic extraction
Kexin Hou, Xuebing Yang, Meili Bao, Fengli Chen, Hao Tian, Lei Yang ChemistryIndustrial Crops and Products2018
Evaluation of cultivars diversity and lipid composition properties of Idesia polycarpa var. vestita Diels
Leyan Wen, Xuwen Xiang, Zhirong Wang, Qingqing Yang, Zehang Guo, Pimiao Huang, Jianmei Mao, Xiaofeng An, Jianquan Kan
First published: 21 August 2022
https://doi.org/10.1111/1750-3841.16293
4. Fruit quality of 12 provenances of Idesia polycarpa in China. DOI: https://doi.org/10.1234/4.2014.5241Author(s):Li Dai, Yanmei Wang, Zhen Liu *, Fei Li, Haiyang WangRecieved Date: 2014-01-10, Accepted Date: 2014-03-30
2. Materials and Methods
Lines 78-84, 2.2. Preparation of IPPO
Please indicate:
-initial moisture content of the fruit, its initial average weight,
-indicate % (average) of pulp and the seeds
-Moisture content after dehydration,
-How was the dehydration is performed, T, duration, in dark????
-Its maturity index, or stage
-Temperature of the extraction (cold Folch extraction method is performed???)
-lines 85 2.3. Pulp content, oil content and physicochemical property, correct properties
-No density, viscosity, boiling point, measurements are done for these oils, so only chemical properties and stability indexes were analyzed. The manuscript title, abstract should be revised accordingly
-Lines 90-96
2.4. Fatty acid composition
GC is it quantitative ?, if yes, standard used to be mentioned, results of fatty acids are expressed in % peak area % in total oil ; if analyses are qualitative, this reduce the scientific quality of the manuscript,
Please indicate if analytical method allows distinguishing trans and cis fatty acids?
In paragraphs 2.5, 2.6 and 2.7, the calibration curve and min-max concentrations of standards should be indicated.
-Principal components analysis may be useful to more explain variabilities between measured parameters of IPPO from five regions. according to different regions, Yi may be, SFA, UFA, Contents of tocopherols, β-sitosterol, and polyphenols, the four free radical scavenging abilities, acid value and peroxide value
Results and discussion
-Lines 2015-225, the percentages car earea of the peak or g /100g
-Table 2. Fatty acid content and composition of IPPO of five regions. Unit of fatty acids should be added, number of sample analyzed. In tables footnotes and in material section.
Are there any omega 6 fatty acid, omega 3/omega 6 ratio could be calculated and compared to other vegetables oils,
If there aren’t omega 6 fatty acid, the ratio of SFE/Unsaturated fatty acid may be calculated and discussed in comparison with other vegetables oils
-number of analyzed samples and units should be added in all tables title and footnotes
-table 2 showed as reported 6 main fatty acids, but total SFA +UFA = 100% so, all amino acids are quantified, or chromatograms shows only six fatty acids.
-a GC chromatogram for each oil varieties maybe added in supplemented file or in the main Manuscript,
· - Conclusion
· Conclusion should have performed after deep statistical analysis of obtained results such as PCA analysis of
Reviewer 3 Report
Manuscript entitled “Analysis of chemical composition and antioxidant activity of Idesia polycarpa pulp oil from five regions in China” is very poor prepared, content a lot of mistakes, errors and have very low scientific level. It is only report about one more unconventional plant oil extracted from Idesia polycarpa.
The English language is very poor and many editorial errors like the lack of spaces between words.
In manuscript is lack of method for PV, AV, IV and saponification value determination. It is therefore difficult to assess the results obtained.
The level of tocopherols in the oil is strongly overestimated and the determination of sitosterol is questionable due to the lack of derivatisation of the sample and the absence of an internal standard.
In Table 2 the linoleic acid is calculated as MUFA!!!
What is palm oil acid.
Reviewer 4 Report
Dear Authors,
I had the opportunity to review your article. The topic seems to be new and interesting, the methods of analysis are appropriate. Nevertheless, the manuscript needs some corrections. Below comment on its content.
Comment 1: Please remove unnecessary spaces before and after “-“, and “±” in the manuscript.
Comment 2: Please add the full name of IPPO in the introduction.
Comment 3: Line 66: Please add or remove space where needed.
Comment 4: Please add how the pulp was dried.
Comment 5: Please add in section 2.3 all numbers of food AOCS method for each analysis.
Comment 6: Please add information in section 2.4 about the volume of injection, how you identified fatty acids, results were presented quantitatively or qualitatively. What type of gases were used, carrying and makeup? Same thing for other chromatographic methods.
Comment 7: Line 101: Please start giving information about column size, starting with the length.
Comment 8: Line 118: Please give information about the solution used for the calibration curve for each method used to analyze AA.
Comment 9: Please remove unnecessary spaces in the Results and discussion section.
Comment 10: Table 1: why in the case of “Ningqiang”, the content of pulp and oil gives a higher result than 100%?
Comment 11: p-value should be in lowercase.
Comment 12: line 188: Table use lowercase.
Comment 13: Table 4: Please add space between “ABTS (umol TE/100g)”.
Comment 14: Please change the manuscript umol for µmol. I think this is a suitable unit.
Comment 15: In the text, if we are talking about DPPH or ABTS radicals, such information should be marked as DPPH* and ABTS *+
Round 2
Reviewer 2 Report
Reviewing of Analysis of chemical composition and antioxidant activity of Idesia polycarpa pulp oil from five regions in China
Reference. ID: foods-2204733
Second evaluation done on 28-02-2023
Mains serious concerns in revised manuscript:
1. -lack of spaces between may word, lumber and units throughout the MS.
2. -Improvement of English is required in all added sections in revised manuscript,
3. -PCA is not performed appropriately, it should be explained in material and method, data should be treated with expert in PCA and the main biplots could be generated and discussed.
4. Other remarks are below:
Line 125: Please check is this ref added in the revised MS AOCS Am 2-93 is also added in reference list
Line 131 space is missing: 50mg
line 140 and line 169: Only significant number are required Helium (99.999% purity) !!!
Line 183, sentence to be reformulated !!retention time of the standards in the chromatogram.
2.9. Data analysis 239-242: the authors should add how PCA was performed.
Line 317: reformulate and think this info in unnecessary:
the contents of the three fatty acids were found to be below 1.0% and were 317 not considered as functional ingredients.
Table 2, add (%) in the head of the column and not each fatty acid
Line 603 the original date., data or date, check the whole paragraph for English and mistakes
Line 603: what does mean indicators?? you want to tell oil samples, please check ???"scatter diagram of 22 indicators."
There are issues with PCA analysis
firstly, the PCA analysis should be presented in 2.9. Data analysis 239-242: Line 598 to 629, and rewritten in 3.7 Principal component analysis, to be checked and rewritten. It seems to not be done appropriately,
The have to ask specialist in statistical analysis to perform PCA in generated data, define the observations (are oil samples) and the responses (measured variables): two plots are generated: PCA biplot of objects and component loads for the grouping of descriptors / measured responses (a) and groups of oil samples (b). Another table of Pearson correlation between variables could be interesting to discuss but table 5 Table 5 is unnecessary and it indicate that PCA is not performed appropriately.
Questions to my first revision were generally responded but added sections poses some problem as presented abovee

Reviewer 3 Report
In the table 2 the "content" of fatty acids must be delete. The Authors showed only percentage composition!
Author Response
Comment 1: In the table 2 the "content" of fatty acids must be delete. The Authors showed only percentage composition!
Response: Thanks for your careful comment. The “content” has been deleted as suggested. Thank you again for your kind work.

Reviewer 4 Report
Dear Authors,
Thank you for submitting your manuscript. I am pleased to inform you that I accept your corrections.
Author Response
Thanks very much for your kind work and consideration on revision of our paper. On behalf of my co-authors, we would like to express our great appreciation to you.